# Optimization and evaluation of a two-stage chromogenic assay procedure for measurement of emicizumab plasma levels

**Nasim Shahidi Hamedani** *, **Johannes Oldenburg, Bernd Pötzsch, Jens Müller**

Institute of Experimental Hematology and Transfusion Medicine, University Hospital Bonn, Bonn, Germany

* nasim.shahidi_hamedani@ukbonn.de

## Abstract

Emicizumab mimics the hemostatic activity of activated factor VIII (FVIIIa) within the tenase complex. Despite functional similarities between FVIIIa and emicizumab, conventional laboratory methods designed for monitoring of FVIII activity are inappropriate for the measurement of emicizumab. At present, a modified one stage (FVIII) assay (mOSA) is mainly used for emicizumab monitoring. Two-stage chromogenic FVIII assays based on human factors can be used, although limited performance due to lack of corresponding optimization might be observed. Furthermore, the presence of FVIII or anticoagulants in the patient sample may falsify assay results. To address these issues, we optimized and evaluated a two-stage chromogenic assay (emi-tenase) for measurement of emicizumab in plasma samples. Heat inactivation of samples was established to abolish the influence of endogenous or substituted FVIII. The lower limit of quantification (LLoQ) was found to be 2 µg/ml in a manual assay format and 9.5 µg/ml on an automated coagulation analyzer. Intra- and inter-assay coefficients of variation (CV) did not exceed 20%. Analysis of 17 patient plasma samples with severe haemophilia A under emicizumab treatment showed good correlation of results between the emi-tenase assay and the mOSA (Cohens Kappa coefficient = 0.9). Taken together, the emi-tenase assay allows specific measurement of emicizumab plasma levels over a broad concentration range (10 µg/ml to 100 µg/ml). The assay can be applied on an automated coagulation analyzer, demonstrating its applicability within a routine laboratory setting.

## Introduction

Routine administration of exogenous factor VIII (FVIII) for bleeding prophylaxis in patients with haemophilia A (HA) improves patients' quality of life by reducing bleeding episodes and bleeding-related complications [1, 2]. However, 25 to 30% of patients with severe HA and receiving FVIII replacement therapy develop alloantibodies to FVIII and therefore show spontaneous and traumatic bleeding episodes [3]. For HA patients developing autoantibodies against FVIII, on-demand treatment or regular prophylaxis with bypassing agents (BPAs) such as administration of activated prothrombin complex concentrate (aPCC), recombinant

**Data Availability Statement:** The data set required to replicate the study findings reported in the article

are openly available in Zenodo.org at https://zenodo.org/record/6697795#.YrQxiNNCSUk.

**Funding:** The authors received no specific funding for this work.

**Competing interests:** The authors have declared that no competing interest exist.

activated FVII (rFVIIa), and plasma-derived Factor VIIa/Factor X complex are available or under development [4–7]. From 2017, the bioengineered antibody named emicizumab (Hemlibra®, F Hoffmann-La Roche, Basel, Switzerland) has been approved for routine prophylaxis to prevent or reduce the frequency of bleeding episodes in adults and children with HA with or without inhibitors [8–11].

Emicizumab mimics the procoagulant function of activated (co)factor VIII [FVIIIa] by simultaneous binding to (activated) factors IXa (FIX) and X (FX) [12, 13]. It is therefore able to take over the tasks executed by FVIIIa, albeit at lower turnover rates. The catalytic efficiency ($k_{cat}/K_m$) by which FVIIIa enhances activated factor X (FXa) generation is 10 times higher than that of emicizumab while patients under prophylactic treatment with emicizumab show annualized bleeding rates and hemostatic potentials close to that of patients with moderate hemophilia [14, 15]. In the HAVEN 4 cohort study, a steady state plasma concentration of approximately 40 μg/ml emicizumab was achieved 24 weeks after initiation of an every 4-week treatment scheme [16]. In addition, in a phase 3 study of emicizumab prophylaxis in children aged <12 years with HA with inhibitors, mean steady-state concentrations were maintained at effective levels of 38–50 μg/ml [17].

Plasma levels of emicizumab can be measured directly or indirectly. Direct methods measure emicizumab mostly through immunological methods while indirect methods quantify the functional procoagulant potential of emicizumab [12, 13]. Moreover, immunological methods are in general hampered by long turn-around times while functional testing is less time-consuming and also well established in monitoring of FVIII replacement approaches and monitoring of anticoagulant drugs [18].

Previous studies showed that emicizumab plasma levels can be measured by a modified FVIII one-stage assay (mOSA), a two-stage chromogenic FVIII-assay or can be estimated by subjecting prediluted plasma samples to a standard FVIII one-stage assay (sOSA) [12, 19, 20]. The mOSA is basically a FVIII-activity assay that operates at higher sample predilutions and the use of emicizumab calibrators for the construction of a corresponding standard curve [18, 21]. The mOSA method was shown to have good precision and reproducibility and emicizumab levels determined using the mOSA correlated well with the ELISA method previously used in the HAVEN clinical trials [21, 22].

In a recently published study, pre-diluted plasma samples were subjected to the sOSA and the calculated FVIII activity was used for emicizumab plasma level approximation (ELA), resembling the emicizumab level in μg/ml of the undiluted sample. Although this method demonstrated substantial agreement with the mOSA method, increased variation of ELA was observed at emicizumab levels above 30 μg/ml. Hence this assay is useful and therefore was proposed for discrimination between sub-therapeutic and therapeutic emicizumab plasma levels [20].

A FVIII two-stage chromogenic substrate assay (CSA) which uses human activated Factor IX (FIXa) and FX is responsive to emicizumab and is therefore, in principle, operational to measure emicizumab in patient plasma samples. However, currently available assays were developed and optimized for the measurement of plasma FVIII levels while optimal assay conditions for the measurement of emicizumab plasma levels using a CSA have not been determined yet [23].

Here, we present data on the optimization of a two-stage CSA for accurate and reliable quantification of emicizumab plasma levels. We also address pre-analytical heat-inactivation of samples to eliminate FVIII that may interfere with emicizumab assay results [24]. The assay was also established on an automated coagulation analyzer to evaluate its applicability in a routine laboratory setting. A schematic illustration of the two-stage chromogenic assay is presented in Fig 1.

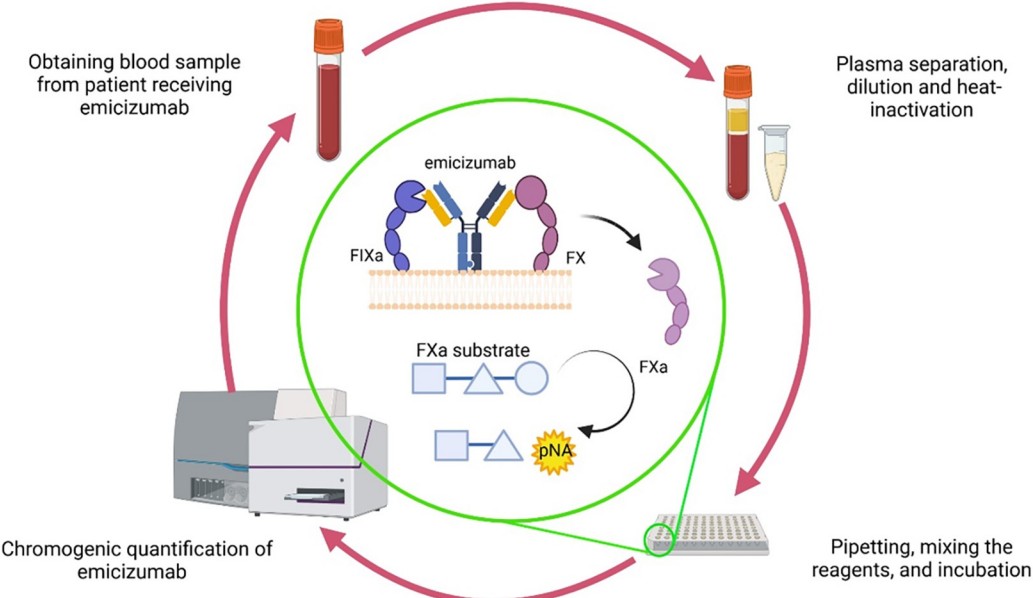

**Fig 1. Principle of the two-stage chromogenic assay procedure for quantification of emicizumab plasma levels.**

## Materials and methods

### Chemicals and reagents

Tris-HCl, NaCl, $CaCl_2$, $MgCl_2$, and bovine serum albumin (BSA) were purchased from Merck/Sigma-Aldrich (Darmstadt, Germany). Plasma-purified human activated factor IX (FIXa), human factor IX (FIX), human activated factor X (FXa), and human factor X (FX) were purchased from Haematologic Technologies Inc. (Essex Junction, USA). The FXa-chromogenic substrate Z-D-Arg-Gly-Arg-pNA·2HCl (S-2765) was purchased from Chromogenix (Milan, Italy). Factor VIII-deficient plasma, standard human plasma (SHP), Actin FS, and calcium chloride solution (25 mM) were obtained from Siemens Healthcare Diagnostics (Siemens, Marburg, Germany). Phospholipids (a mixture of phosphatidyl choline, phosphatidyl serine and sphingomyelin) were purchased from Rossix (Mölndal, Sweden). Emicizumab plasma calibrator and controls were purchased from $r^2$ Diagnostics (South Bend USA).

### Plasma samples

Residual plasma samples from 17 adult and adolescent HA outpatients developing inhibitors against FVIII and receiving subcutaneous injection of emicizumab were included in the present study. Samples from patients showing a FVIII activity above 2% were excluded. All patients were registered in the Bonn Hemophilia Center and analysis was done in the context of an extended diagnostic work-up. Normal human plasma was obtained from healthy donors and all participants gave written informed consent as approved by the local institutional review board (University of Bonn. Ethics Committee; No. 070/05). Plasma was obtained by centrifugation of blood samples at 2,500 x g for 15 min and aliquots stored at ≤ -40°C until analyzed.

### Modified FVIII one-stage clotting assay for determination of emicizumab plasma levels

The applied modified FVIII one-stage clotting assay (mOSA) for emicizumab measurements was based on the activated partial thromboplastin time (aPTT) with the pre-diluted patient

plasma sample further diluted in FVIII-deficient plasma. In detail, patient plasma was pre-diluted 1:8 in 0.9% (w/v) NaCl solution and then further diluted 1:10 in FVIII-deficient plasma in a total volume of 50 μl. Then, 50 μl of the aPTT-reagent Actin FS were added and incubated for 3 min at 37˚C. Subsequently, the clotting process was initiated by addition of 50 μl of 25 mM CaCl$_2$ solution. The assay was calibrated and controlled against emicizumab (r$^2$ Diagnostics, South Bend, IN, USA) and all analysis was performed using the automated BCS XP coagulation analyzer (Siemens).

## Optimization of the chromogenic emicizumab assay

A two-stage tenase assay configuration was designed and optimized to measure emicizumab plasma levels. Plasma samples were subjected to heat inactivation at 56˚C for 30 minutes followed by centrifugation. The final assay volume was set to 100 μl in the manual format for assay optimization. The general assay setting and pipetting steps were defined as follows: 25 μl of (prediluted) plasma samples, calibrators or controls, 25 μl of the FIXa reagent, and 25 μl of the FX reagent. After an incubation step, 25 μl of the FXa-chromogenic substrate S-2765 were added and the kinetics of substrate hydrolysis measured for 5 minutes at 405 nm using a microplate reader (Synergy 2, Biotek, Bad Friedrichshall, Germany). All samples and reagents were diluted in assay buffer (20 mM Tris-HCl, 137 mM NaCl, 5 mM CaCl$_2$, 1 mM MgCl$_2$, 1 mg/ml BSA and 20 μg/ml phospholipids (PLs), pH 7.4).

A BCS XP coagulation analyzer (Siemens Healthineers) was used for automation of the optimized assay scaled-up to 200 μl/reaction. Accordingly, samples, calibrators, or controls were pre-diluted 1:20 in 0.9% NaCl solution containing 5 mM of the anti-polymerant peptide Gly-Pro-Arg-Pro (GPRP). Afterwards, 50 μl of the diluted samples were mixed with 50 μl each of FIXa-reagent (160 nM) and FX-reagent (600 nM) that placed on separate cooling positions of the BCS XP. After an incubation step of 5 min, 50 μl of S-2765 were added and the kinetics of substrate conversion measured for 5 minutes.

Both, the manual and the automated assay were calibrated and controlled by using the above described emicizumab plasma calibrators and controls.

## Evaluation of the chromogenic emicizumab assay

To evaluate quantitative assay performance, the recovery, the limit of detection (LoD), the lower limit of quantification (LLoQ), intra-assay variability (repeatability), and inter-assay variability (reproducibility) were determined [25, 26]. To test for recovery, plasma samples obtained from three healthy donors and from three patients with severe HA (FVIII activity < 2%) were spiked with 20, 40, and 80 μg/ml emicizumab and tested under optimized assay conditions. To determine the LoD and LLoQ of the assays, emicizumab plasma concentrations of 1 to 100 μg/ml and six replicates of the blank sample were analyzed. The LoD and LLoQ were defined as the concentrations that correspond to the signals of the blanks + three or nine times the standard deviation (SD), respectively. To determine within- and day-to-day assay variability, repeated measurements of plasma samples taken from five emicizumab-treated hemophilia patients were performed.

## Statistics

All samples were analyzed in triplicates if not otherwise stated. Statistical analysis and graphical presentation were performed using GraphPad Prism version 8.3.0 for Windows (GraphPad Software, La Jolla California USA, www.graphpad.com). Best-fit interpolation was applied for regression analysis. Bland Altman plots were constructed by plotting the differences against the mean results of the two methods [27].

## Results

### Basic chromogenic assay optimization in the purified system

Emicizumab-dependent formation of FXa was monitored through hydrolysis rates of the chromogenic FXa-specific substrate S-2765. To avoid a negative impact of limited availability of S-2765 on assay performance, the dose-relationship of FXa-mediated S-2765 hydrolysis was tested using increasing concentrations of FXa and a final S-2765 concentration of 0.5 mM (2 times the $K_M$ value). A log-linear rate of substrate hydrolysis was observed up to a FXa-concentration of approx. 15 nM in the final mixture (Fig 2A). Consequently, a preliminary final FX-concentration 10 times higher (150 nM) than the assessed maximum concentration of FXa was used in the following experiments.

The formation of the emicizumab-mediated tenase complex requires FIXa. In order to optimize the FIXa-concentration, increasing concentrations of FIXa were analyzed in conjunction with 150 nM FX and 20 µg/ml PLs (final concentrations) in the absence or presence of emicizumab (0 µg/ml, 0.25 µg/ml or 2.5 µg/ml within the final reaction mixture, corresponding to 0 µg/ml, 10 µg/ml [low] or 100 µg/ml [high] concentration within the original sample regarding the 1:10 preliminary dilution factor of samples and 1:4 dilution caused by addition of

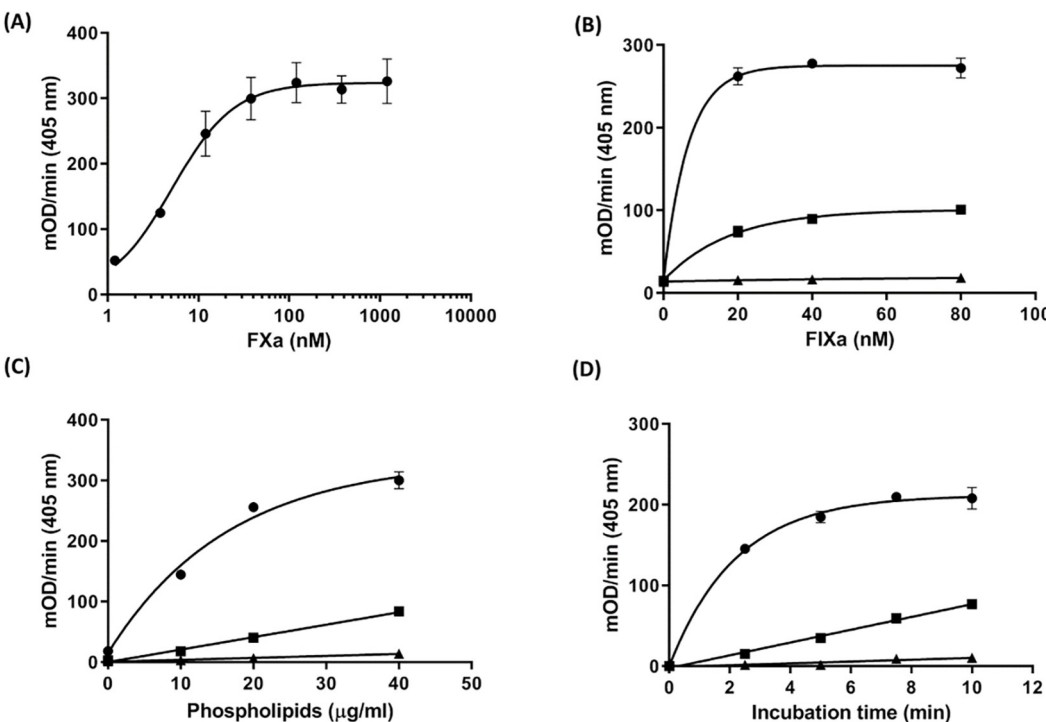

**Fig 2. Assay optimization.** (A) FXa-catalyzed substrate hydrolysis. To evaluate the maximal acceptable FXa concentration, 75 µl of FXa diluted in a Tris-based buffer containing 20 µg/ml phospholipids and 5 mM CaCl$_2$ were mixed with 25 µl of 2 µM FXa-specific chromogenic substrate S-2765 and substrate conversion was measured. (B) Optimization of the FIXa concentration. Different concentrations of FIXa (25 µl) were mixed with 25 µl of different concentrations of emicizumab and 25 µl FX (final concentration of 150 nM) in the presence of phospholipids and calcium. After 5 min incubation at RT, 25 µl of S-2765 (2 mM) were added and the substrate hydrolysis was measured. (C) Evaluation of the phospholipid concentration. The influence of increasing concentrations of phospholipids on emicizumab dependent FXa formation was measured at FIXa (40 nM) and FX (150 nM) in a Tris-based buffer containing different levels of phospholipids. (D) Influence of the incubation time on FXa formation. A reaction mixture prepared by mixing of FIXa (40 nM), FX (150 nM) and different concentrations of emicizumab was incubated for various incubation time followed by addition of S-2765 substrate and FXa-formation measurement. All samples were measured at 405 nm using the Synergy2 microplate reader. All data were collected in triplicate and expressed as the means ± s.d. In figures B to D, ● indicates 100 µg/ml emicizumab; ■, 10 µg/ml emicizumab and ▲indicated without emicizumab.

reagents to the pre-diluted sample). The amount of FXa formed was monitored through S-2765 cleavage after a preliminary incubation time of 5 min. Up to a final FIXa-concentration of 80 nM, no detectable activation of FX in the absence of emicizumab was observed, while increasing concentrations of emicizumab led to expected substrate conversion rates (Fig 2B). With respect to both, assay sensitivity and the required dynamic range, a final concentration of 40 nM was found to be optimal for FIXa. In order to further evaluate the aforementioned findings, additional experiments for the assessment of different combinations of FX- and FIXa-concentrations were applied. As shown in S1 Fig, indeed, final concentrations of 150 nM FX and 40 nM FIXa were found to be optimal und therefore used for all further experiments.

Although emicizumab is neither able to bind to PLs nor enhances the binding of FIX(a) or FX to the PLs surface, the presence of PLs in the reaction mixtures is needed to allow for optimal alignment of FIXa/FX for emicizumab binding [14]. To evaluate the optimal PLs-concentration, emicizumab-dependent FXa formation was measured at increasing concentrations of PLs under so far optimized assay conditions. Increasing rates of FXa formation were observed at low and high emicizumab concentrations with increasing PLs concentrations. However, a final concentration of 20 μg/ml was confirmed to facilitate both, adequate assay sensitivity and an appropriate dynamic range (Fig 2C).

In order to assess the influence of different incubation times on assay sensitivity, FXa generation was continued for up to 10 min before addition of the chromogenic substrate (Fig 2D). A linear increase in FXa formation over assessed incubation times was observed for the tested low emicizumab concentration. However, since FXa formation rates at the high emicizumab concentration peaked at incubation times of > 5 min, the initially applied, preliminary incubation time of 5 min was maintained during all further experiments.

## Chromogenic assay optimization for analysis of patients' plasma samples

Endogenous or substituted FVIII increases the tenase activity and, therefore, if present in the sample, diminishes the emicizumab-specificity of the assay. Since FVIII is highly sensitive to heat-inactivation, the influence of pre-analytical heat-inactivation on FVIII- and emicizumab-activity was tested in the plasma matrix at an initial 1 in 10 dilution and the presence (5 mM) of the clot inhibitor peptide Gly-Pro-Arg-Pro (GPRP) which mimics the N-terminal Gly-Pro-Arg region in the α chain of fibrinogen [28]. Data showed that heat inactivation effectively reduced endogenous FVIII activity without significant influence on emicizumab quantification (Fig 3A). In addition, applying heat inactivation to FVIII-deficient plasma samples spiked with different concentrations of emicizumab also led to virtually higher detection sensitivity (Fig 3B). While this effect may be due to the elimination of further disturbing substances in the plasma samples by heat, the results demonstrate that heat-inactivation of samples appears to be an easy to apply but also effective measure to facilitate the specific measurement of plasma emicizumab levels.

To further evaluate the influence of the plasma matrix on assay performance and to determine the optimal dilution factor, heat-inactivated, FVIII-deficient plasma or assay buffer was spiked with emicizumab to achieve final concentrations of 0 μg/ml, 10 μg/ml and 100 μg/ml. Subsequently, samples were serially diluted in assay buffer containing 5 mM GPRP and introduced to the optimized assay setting as described above. As shown in Fig 3C and 3D, FXa generation rates comparable between spiked plasma and buffer samples were achieved from dilution of 1:20 that, accordingly, was applied for all further experiments.

## The effect of high plasma concentrations of FIX and FX on assay results

Emicizumab shows comparable binding affinity to FIX/FIXa as well as to FX/FXa [29]. Hence, endogenous FIX or FX might interfere with the determination of emicizumab plasma levels.

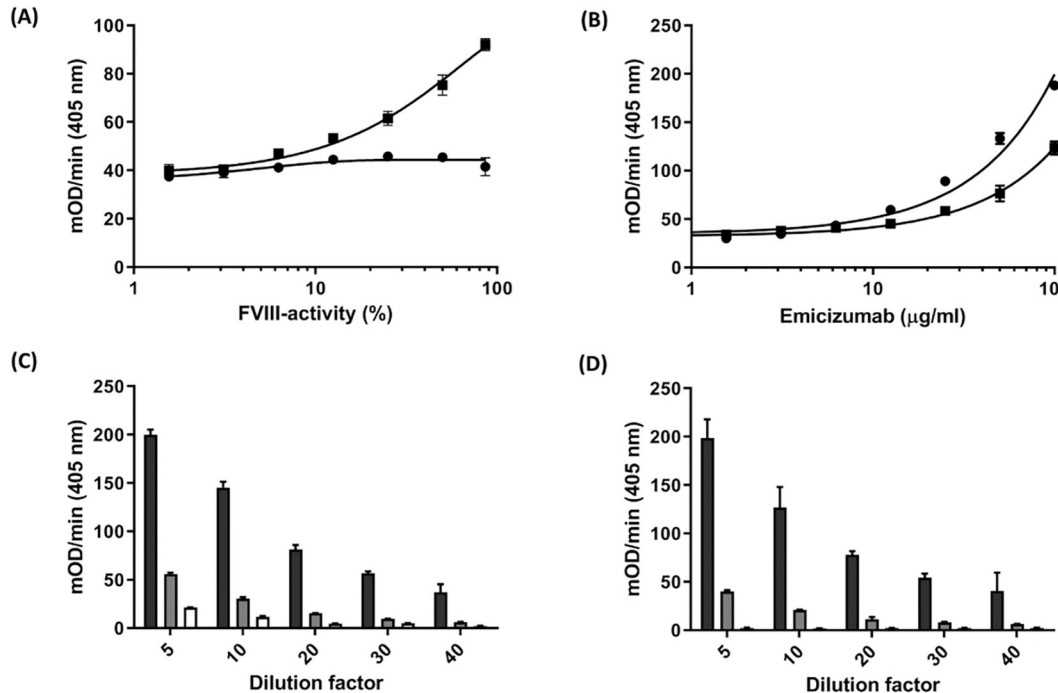

**Fig 3. Pre-analytical inactivation of endogenous factor VIII and minimizing the impact of plasma matrix on determination of emicizumab plasma levels.** (A) FVIII-deficient plasma was mixed with standard plasma to reach various activity levels of FVIII and then spiked with emicizumab (10 μg/ml) and subjected to emi-tenase assay with (●) or without (■) heat inactivation at 56°C for 40 min. Plasma samples were diluted with NaCl 0.9% and FXa generation was measured after addition of FIXa (40 nM), FX (150 nM) and, S-2765 (500 nM). (B) The impact of heat inactivation on the measurement sensitivity using emicizumab spiked FVIII-deficient plasma. FVIII-deficient plasma was spiked with increasing concentrations of emicizumab and subjected to the same test system with (●) and without (■) heat inactivation. To find out the optimum plasma dilution, citrated FVIII-deficient plasma (C) or NaCl 0.9% containing 0.1% BSA (D) were spiked with emicizumab (100 μg/ml, black bars; 10 μg/ml, grey bars and without emicizumab, white bars) and were heat inactivated. Samples were diluted with NaCl 0.9% to reach different dilution factors and incubated with FIXa (40 nM) and FX (150 nM) for 5 min. FXa generation was measured subsequent to addition of S-2765 chromogenic substrate (500 μM). In all experiments, substrate conversion was measured at 405 nm using microtiter plate reader. Data were collected in triplicate in a kinetic measurement of 5 min and expressed as the means ± s.d.

To determine the impact of different and especially elevated plasma levels of FIX and FX on assay performance, FVIII-deficient plasma was spiked with FIX or FX to, in addition to 100% (approx. 90 nM FIX or 170 nM FX), reaching to 150% and 200% of normal plasma concentrations. Then samples were spiked with low and high doses of emicizumab and subjected to the assay. As shown in Fig 4, there was only limited impact of excess amounts of FIX or FX on assay performance. However, a slight increase in FXa generation and, thus, determined plasma emicizumab, was observed at high levels of FX when 10 μg/ml emicizumab was used as the input concentration although this effect was less pronounced by using 100 μg/ml emicizumab.

## Evaluation and automation of the optimized chromogenic assay (emi-tenase)

Evaluation of quantitative assay performance included determination of the lowest analyte concentration likely to be reliably distinguished from the blank values (the limit of detection, LoD) and the concentration at which quantitative detection is feasible (the lower limit of quantification, LLoQ). Further evaluation included the recovery and the reproducibility of the assay. Regarding the above-described microtiter-based manual assay format, a typical standard

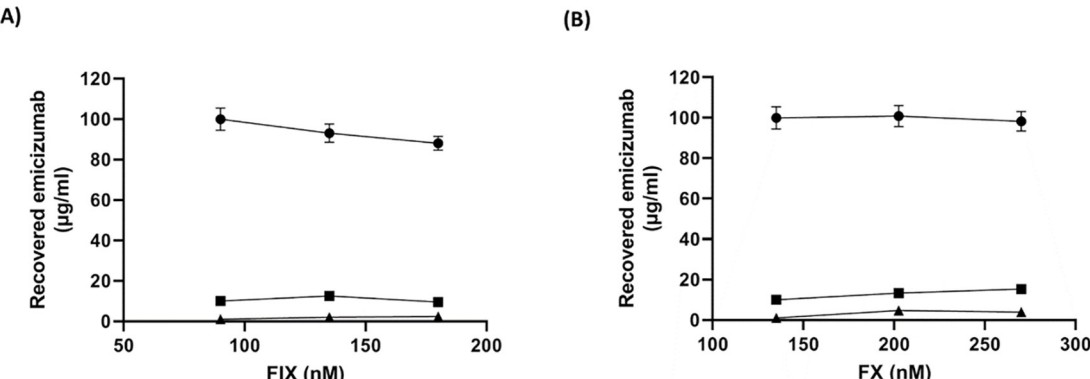

**Fig 4. The impact of higher concentrations of FIX and FX on determination of emicizumab plasma level.** FVIII-deficient plasma was spiked with FIX (A) or FX (B) to reach 100%, 150% and 200% of normal plasma levels in molarity. Then samples were spiked with emicizumab (● indicates 100 μg/ml emicizumab; ■ 10 μg/ml emicizumab and ▲ without emicizumab). A dilution factor of 1:20 of each plasma samples was incubated with FIXa (40 nM) and FX (150 nM) for 5 min followed by addition of S-2765 substrate (500 nM final concentration). Substrate conversion was measured at 405 nm using microtiter plate reader. Data were collected in triplicate in a kinetic measurement of 5 min and expressed as the means ± s.d.

curve is shown in S2 Fig. The LoD and LLoQ of the manual assay, which were defined as three and nine times the standard deviation (SD) above the mean of six replicates of the blank were found to be 0.63 μg/ml and 1.96 μg/ml, respectively. Using the BCS XP system for assay automation however, increased the LoD and LLoQ to 6.33 μg/ml and 9.53 μg/ml, respectively (see S2 Fig for standard curve). The intra-assay and inter-assay coefficient of variations (CVs) were calculated using citrated plasma obtained from five patients diagnosed with severe HA (FVIII activity < 2%) and under treatment with emicizumab. For the manual method, the data showed intra-assay CVs that ranged from 4.98% to 7.75%, while inter-assay CVs varied between 9.55% and 18.55%. Performing the assay on the automated coagulometer analyzer demonstrated intra- and inter-assay CVs of 4.41% to 11.95% and 5.93% to 8.38%, respectively (Table 1). Mean recoveries for low (20 μg/ml), medium (40 μg/ml) and high (80 μg/ml) emicizumab levels were 91.5%, 90% and 95.8%, respectively (Table 2).

## Chromogenic assay (emi-tenase) vs. mOSC: Comparison of assay results

To assess the agreement between the mOSC and the newly developed chromogenic emicizumab assay (emi-tenase), correlation- as well as Bland-Altman analysis was applied [27]. Briefly,

**Table 1. Intra-assay and inter-assay precision.**

|  | Manual format | | | | Automated | | | |
|  | Intra-assay | | Inter-assay | | Intra-assay | | Inter-assay | |
|  | Mean | CV*(%) | Mean | CV(%) | Mean | CV(%) | Mean | CV(%) |
|---|---|---|---|---|---|---|---|---|
| Patient 1 | 42.73 | 6.5 | 39.61 | 18.55 | 48.09 | 9.24 | 48.05 | 7.36 |
| Patient 2 | 30,81 | 6.65 | 28,56 | 14.23 | 33.52 | 11.39 | 32.76 | 7.77 |
| Patient 3 | 39.17 | 6.89 | 45.41 | 9.55 | 55.82 | 7.9 | 56.13 | 8.38 |
| Patient 4 | 43.46 | 7.75 | 37.26 | 14.98 | 31.21 | 4.41 | 30.05 | 5.93 |
| Patient 5 | 67.7 | 4.98 | 59.35 | 12.97 | 61.55 | 7.9 | 60.52 | 6.64 |

Emicizumab plasma level was measured manually and automatically using BCS-XP system in citrated plasma obtained from five patients with haemophilia A receiving emicizumab.

* CV, coefficient of variation

**Table 2. Spike and recovery analysis.**

| | Spike Level | | |
|---|---|---|---|
| Sample | Low spike (20 µg/ml) | Medium spike (40 µg/ml) | High spike (80 µg/ml) |
| Donor 1 | 11.87 | 35.53 | 64.32 |
| Donor 2 | 21.87 | 45.69 | 100.87 |
| Donor 3 | 17.59 | 31.28 | 75.12 |
| Patient 1 | 17.99 | 33.03 | 80.24 |
| Patient 2 | 19.32 | 36.09 | 69.29 |
| Patient 3 | 21.29 | 34.39 | 69.84 |
| Mean Recovery ± SD | 18.31 ± 3.58 | 36 ± 4.93 | 76.61 ± 13.74 |
| Mean Recovery ± SD (%) | 91.53 ± 17.9 | 90 ± 12.34 | 95.77 ± 17.18 |

Emicizumab was spiked in citrated plasma obtained from three healthy individual donors and three patients with severe hemophilia A to reach low, medium and high levels of emicizumab. Mean recovery was calculated as a ratio of measured value and spiked value in percentage. All values represent the average of three replicates.

17 plasma samples obtained from HA patients treated with emicizumab were subjected to both assays. In general, a high correlation of results was found, whereat absolute differences did not exceed 10 µg/ml in 16 out of the 17 samples. Overall, however, relative differences of results did not exceed 25% (Fig 5).

## Discussion

Since emicizumab treatment adjusted to body weight usually results in a predictable procoagulant response, drug level monitoring is deemed not to be required during routine prophylaxis. However, there may be a need for accurate measurement of emicizumab plasma levels, e.g. in order to confirm achievement of steady-state target concentrations or in case of unexpected bleeding complications [20, 23].

Although most laboratories have access to the appropriate instrumentation to perform chromogenic assays, some laboratory scientists and clinicians are not familiar with this kind of assays [30]. Furthermore, performing chromogenic assays might suppose to be more expensive

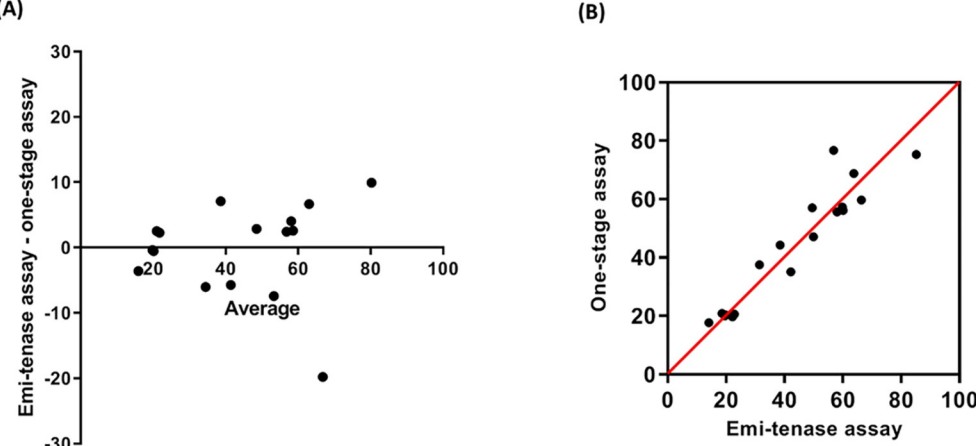

**Fig 5. Method comparison of two measurements.** Seventeen plasma samples from hemophilia patients receiving emicizumab were subjected to APTT-based one-stage assay and Emi-tenase assay. (A) The Bland-Altman agreement analysis was done by plotting the difference of the two paired measurements (Emi-tenase assay–one-stage assay) against the mean of those two measurements. (B) A scatter plot of the relationship between measured values of the same samples using both methods.

than one-stage assays. However, a computer-based model analyzed and compared the costs associated with FVIII OSA and chromogenic assays, concluded that efficient use of reagents (e.g. by aliquoting and freezing prior to use) can render the cost of the OSA and CSA comparable for both single and batch samples [31]. Finally, the CSA does not use FVIII-deficient plasma which is a significant variable and is less prone to interference by some pre-analytical variables such as assay components and plasma matrix [32]. In the present study, (pre)analytical conditions of a chromogenic two-stage assay for emicizumab (emi-tenase) that allow to measure emicizumab over a range from 10 to 100 μg/ml in human plasma was systematically evaluated. Among the commercially available chromogenic substrates, we selected S-2765 because this substrate, in comparison to others, shows relatively high FXa-specificity, as well as a high turn-over rate ($k_{cat}$ of 240 s$^{-1}$) at low $K_M$ (0.26 mM). However, also S-2765 is not monospecific for FXa and relative reactivity to thrombin, activated factor XI (FXIa), activated protein C (APC), plasmin and single chain tissue plasminogen activator has been reported. Thus, in order to avoid any potential interferences on assay results, we decided to directly add FIXa to the assay reactions rather than use FXIa for activation of endogenous FIX in the patient sample [32]. Furthermore, in contrast to FVIII(a), the functional activity of emicizumab does not depend on upstream enzymatic activation of the molecule [14]. Accordingly, there is no need for addition of thrombin (as FVIII-activator) to the first stage of the reaction, preventing direct or indirect (via activation of protein C) unspecific cleavage of S-2765. These points highlight the advantages of the development of a chromogenic assay for emicizumab from scratch, rather than using an assay which has been optimized for FVIII [33].

The same reasoning can be made with respect to the potential interference of (residual) FVIII-activities in the samples to be analyzed. Indeed, to prevent or during acute bleedings, additional application of FVIII may be needed in HA patients treated with emicizumab [34]. Furthermore, also emicizumab-treatment of patients having moderate or acquired HA is currently discussed or already applied [35, 36]. While specific quantification of FVIII in the presence of emicizumab can be easily done using a chromogenic assay for FVIII that is based on non-human (bovine) factors and is therefore insensitive to emicizumab, FVIII will inevitably interfere with the determination of emicizumab plasma levels when using both, the mOSA or a chromogenic assay format based on human factors [37]. Pre-analytic heat inactivation of patients' plasma (56°C for 30 minutes) is important to eliminate interference caused by endogenous or therapeutic FVIII in the plasma samples. While heat-inactivation of plasma samples is an effective and well established pre-analytical measure during application of the Nijmegen-Bethesda-Assay [38], the influence of heat-inactivation of plasma samples directly added to one- or two-stage assays originally developed for functional assessment of clotting factor-activity has not been systematically investigated yet. In order to ensure emicizumab-specificity of the optimized emi-tenase assay, we decided to include heat-inactivation of samples as an integral part of the assay procedure. Thus, it remains unclear to which extend deviations of test results when compared to applied mOSC must be ascribed to the impact of heat-inactivation. However, since no systematic trends between results were observed, a general effect of heat-inactivation appears to be unlikely. Nevertheless, it cannot be excluded that heat-inactivation will lead to altered test results in individual patient samples.

Previous studies showed that emicizumab binds to FIX/FIXa and FX/FXa with comparable affinities [29, 39]. Thus, to assess the potential interference of excess amounts of FIX or FX in patient plasma samples, FXa formation was monitored in the presence of increasing concentrations of FIX or FX. The data showed that even at plasma levels being two times higher as normal, the impact of both, FIX and FX on (relative) assay results was acceptable (<25%), demonstrating assay robustness.

During assay validation, the mOSA performed by Shinohara et al., using either Actin FS or APTT-SLA as an activator reagent, showed a LLoQ of 10 μg/ml emicizumab [21]. Regarding the two-stage emi-tenase assay described in the present paper, the determined LLoQ was higher for the automated format but did not exceed 9.5 μg/ml [21]. Using patient plasma samples rather than more artificial spiked material, the intra-assay precision of both, the manual and the automated assay format ranged from 5% to 11% while inter-assay precision did not exceed 18.5%. Agreement analysis between the automated emi-tenase and the mOSC-assay performed on 17 plasma samples obtained from patients under emicizumab treatment showed high correlation of results whereat relative differences did not exceed 25%. These results further demonstrate that the emi-tenase assay is applicable for determination of emicizumab in clinical plasma samples.

## Conclusion

A chromogenic assay for determination of emicizumab plasma levels was developed and evaluated. The assay showed both, good performance and agreement of results when compared to the mOSA. Automation of the assay could be realized, enabling its application in a routine setting if validated accordingly.

## Supporting information

**S1 Fig. FIXa /FX titration experiment.** Twenty-five microliter of different concentrations of FIXa were mixed with 25 μl of various concentrations of FX in the presence of 100 μg/ml (A) or 10 μg/ml (B) emicizumab. FXa generation efficiency was measured after addition of 25 μl of the chromogenic substrate S-2765 (500 nM). Substrate conversion was measured at 405 nm using a microtiter plate reader. Data were collected in duplicate and showed as the means ± s. d.
(TIF)

**S2 Fig. Standard curves required for emicizumab quantification.** Typical standard curves resulted from different emicizumab concentrations in (A) manual and (B) automated format. Data were collected for manual configuration in triplicate and for automated configuration in duplicate showed as the means ± s.d.
(TIF)

## Acknowledgments

The authors thank Simone Gasper for expert technical assistance and Sneha Singh for preparation of the graphical illustration.

## Author Contributions

**Conceptualization:** Nasim Shahidi Hamedani, Bernd Pötzsch, Jens Müller.

**Investigation:** Nasim Shahidi Hamedani.

**Methodology:** Nasim Shahidi Hamedani, Jens Müller.

**Software:** Nasim Shahidi Hamedani.

**Writing – original draft:** Nasim Shahidi Hamedani, Jens Müller.

**Writing – review & editing:** Nasim Shahidi Hamedani, Johannes Oldenburg, Bernd Pötzsch, Jens Müller.

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
