## [Decision Letter · Decision Letter 0]

20 Jun 2022

PONE-D-22-10383Optimization and evaluation of a two-stage chromogenic assay procedure for measurement of emicizumab plasma levelsPLOS ONE

Dear Dr. Hamedani,

Thank you for submitting your manuscript to PLOS ONE. After careful consideration, we feel that it has merit but does not fully meet PLOS ONE’s publication criteria as it currently stands. Therefore, we invite you to submit a revised version of the manuscript that addresses the points raised during the review process.

We look forward to receiving your revised manuscript.

Kind regards,

Arijit Biswas

Academic Editor

PLOS ONE

Journal Requirements:

2. In your Methods section, please provide additional information on the patients/donors and ensure you have included details on (1) source of patients/donors(2) inclusion/exclusion criteria for recruitment

Please review your reference list to ensure that it is complete and correct. If you have cited papers that have been retracted, please include the rationale for doing so in the manuscript text, or remove these references and replace them with relevant current references. Any changes to the reference list should be mentioned in the rebuttal letter that accompanies your revised manuscript. If you need to cite a retracted article, indicate the article’s retracted status in the References list and also include a citation and full reference for the retraction notic

Reviewers' comments:

Reviewer's Responses to Questions

**Comments to the Author**

1. Is the manuscript technically sound, and do the data support the conclusions?

Reviewer #1: Yes

Reviewer #2: Yes

2. Has the statistical analysis been performed appropriately and rigorously? 

Reviewer #1: Yes

Reviewer #2: Yes

3. Have the authors made all data underlying the findings in their manuscript fully available?

Reviewer #1: Yes

Reviewer #2: Yes

4. Is the manuscript presented in an intelligible fashion and written in standard English?

Reviewer #1: Yes

Reviewer #2: Yes

5. Review Comments to the Author

Reviewer #1: Dear editor of PLOS ONE journal

The manuscript entitled “Optimization and evaluation of a two-stage chromogenic assay procedure for measurement of emicizumab plasma levels” was reviewed. The comments have been listed below:

1- The study has been designed well.

2- Different parts of manuscript have been written appropriately.

3) Which criteria were used for patient selection? Please explain completely.

4) How many healthy blood donors were selected for this study?

Regards

Reviewer #2: 1- The advantage of heat-inactivation should be explained more precisely

2. The authors should explain why the patients‘ samples are too

3. Please specify if the chromogenic FXa method was a kinetic or an endpoint method.

4. What are “general chemicals”. This should be specified when describing the details of

the assays. Accordingly, the first sentence can be eliminated.

6. PLOS authors have the option to publish the peer review history of their article (what does this mean?). If published, this will include your full peer review and any attached files.

Reviewer #1: No

Reviewer #2: No

---

## [Author Response · Author response to Decision Letter 0]

28 Jun 2022

Point-by-point response to reviewers

Submission ID: PONE-D-22-10383

Optimization and evaluation of a two-stage chromogenic assay procedure for measurement of emicizumab plasma levels

Dear Editor,

Here we would like to express our sincere thanks for considering our manuscript to be published in “PLOS ONE”. Please find enclosed a point-to-point response to the reviewers’ comments. We would like to emphasis that the additional journal requirements number 1 to 3 are addressed in the main text.

We hope that the revised version is suitable for publication.

Sincerely yours,

Dr. Nasim Shahidi Hamedani

RESPONSE TO REVIEWERS’ COMMENTS

General authors’ comments:

We would like to thank the reviewers for their critical evaluation of our manuscript. Applying mentioned changes helped us to improve the scientific impact of our manuscript. All the changes are made using tracked changes. 

Comment #1, Reviewer #1: 

The study has been designed well.

Response: We would like to thank the positive evaluation of the reviewer.

Comment #2, Reviewer #1: 

Different parts of manuscript have been written appropriately.

Response: We would like to thank the positive evaluation of the reviewer

Comment #3, Reviewer #1: 

Which criteria were used for patient selection? Please explain completely.

Response: All plasma samples were obtained from adult and adolescent hemophilia A (HA) patients developing inhibitory antibody against FVIII and receiving subcutaneous injection of emicizumab. The patients having the residual activity of FVIII upper than 2% were excluded from the study. The source of the patients/donors was clarified as well. All the changes are indicated in the Material and Methods, Plasma samples section.

Comment #4, Reviewer #1: 

How many healthy blood donors were selected for this study?

Response: Three healthy blood donors were selected for this study and the plasma samples obtained from these donors were subjected to the spike and recovery experiment. The results were included in the Table 2.

Comment #1, Reviewer #2: 

The advantage of heat-inactivation should be explained more precisely.

Response: The heat-inactivation process is described in more details in the material and methods section and its importance is explained precisely in discussion part.

Comment #2, Reviewer #2: 

The authors should explain why the patients‘ samples are too

Response: The comment is not described in full sentence. We estimate it as if the patients’ samples are too small. This is the only limitation of our study that at the time of the study the number of hemophilia patients eligible to receive emicizumab was less in our hemophilia center but according to the Bland-Altman method agreement analysis, the automated emi-tenase and the mOSC-assay which were performed on 17 plasma samples obtained from patients under emicizumab treatment showed high correlation of results whereat relative differences did not exceed 25%. Increasing the sample amount can either improve the agreement between two methods and support more our findings or leave the agreement values the same but not decrease it. Therefore, we are convinced that the number of patients and the design of the study is sufficient to conclude that the two-stage chromogenic assay is in accordance with one-stage clotting assay to quantify the emicizumab plasma levels.

Comment #3, Reviewer #2: 

Please specify if the chromogenic FXa method was a kinetic or an endpoint method.

Response: A kinetic measurement of substrate conversion for 5 minutes is added to the materials and methods section (line 149).

Comment #4, Reviewer #2: 

What are “general chemicals”. This should be specified when describing the details of the assays. Accordingly, the first sentence can be eliminated.

Response: The changes are applied.

---

## [Editor Report · Decision Letter 1]

29 Jun 2022

Optimization and evaluation of a two-stage chromogenic assay procedure for measurement of emicizumab plasma levels

PONE-D-22-10383R1

Dear Dr. Hamedani,

We’re pleased to inform you that your manuscript has been judged scientifically suitable for publication and will be formally accepted for publication once it meets all outstanding technical requirements.

Kind regards,

Arijit Biswas

Academic Editor

PLOS ONE

---

## [Editor Report · Acceptance letter]

5 Jul 2022

PONE-D-22-10383R1 

Optimization and evaluation of a two-stage chromogenic assay procedure for measurement of emicizumab plasma levels

Dear Dr. Hamedani:

I'm pleased to inform you that your manuscript has been deemed suitable for publication in PLOS ONE. Congratulations! Your manuscript is now with our production department. 

Kind regards, 

on behalf of

Dr. Arijit Biswas 

Academic Editor

PLOS ONE